# The Bronchoprotective Effects of Dual Pharmacology, Muscarinic Receptor Antagonist and β_2_ Adrenergic Receptor Agonist Navafenterol in Human Small Airways

**DOI:** 10.3390/cells12020240

**Published:** 2023-01-06

**Authors:** Joseph Antony Jude, Ian Dainty, Nikhil Karmacharya, William Jester, Reynold Panettieri

**Affiliations:** 1Pharmacology & Toxicology, Ernest Mario School of Pharmacy, Rutgers, The State University of New Jersey, Piscataway, NJ 08854, USA; 2Rutgers Institute for Translational Medicine & Science (RITMS), Rutgers, The State University of New Jersey, New Brunswick, NJ 08901, USA; 3Bioscience, Research and Early Development, Respiratory & Immunology, BioPharmaceuticals R&D, AstraZeneca, 431 83 Molndal, Sweden

**Keywords:** human airways, asthma, navafenterol, muscarinic antagonist and β_2_AR agonist, human precision-cut lung slice

## Abstract

Bronchodilators and anti-inflammatory agents are the mainstream treatments in chronic obstructive and pulmonary disease (COPD) and asthma. The combination of β_2_ adrenergic receptor (β_2_AR) agonists and muscarinic antagonists shows superior bronchoprotective effects compared to these agents individually. Navafenterol (AZD8871) is a single-molecule, dual pharmacology agent combining muscarinic antagonist and β_2_AR agonist functions, currently in development as a COPD therapeutic. In precision-cut human lung slices (hPCLS), we investigated the bronchoprotective effect of navafenterol against two non-muscarinic contractile agonists, histamine and thromboxane A_2_ (TxA_2_) analog (U46619). Navafenterol pre-treatment significantly attenuated histamine-induced bronchoconstriction and β_2_AR antagonist propranolol reversed this inhibitory effect. TxA_2_ analog-induced bronchoconstriction was attenuated by navafenterol pre-treatment, albeit to a lesser magnitude than that of histamine-induced bronchoconstriction. Propranolol completely reversed the inhibitory effect of navafenterol on TxA_2_ analog-induced bronchoconstriction. In the presence of histamine or TxA_2_ analog, navafenterol exhibits bronchoprotective effect in human airways and it is primarily mediated by β_2_AR agonism of navafenterol.

## 1. Introduction 

Asthma and COPD are chronic airway disorders characterized by airway hyperresponsiveness, inflammation and remodeling. Bronchodilators are the critical components of the therapeutic management of asthma and COPD. In COPD, the combination of a long-acting muscarinic antagonist (LAMA) and a long-acting β_2_ adrenergic agonist (LABA) elicited bronchodilation superior to that by monotherapy with either of these agents [1,2]. Several small, single molecules with dual action—muscarinic antagonism and β_2_ adrenergic agonism (MABA)—have been developed in the past [3,4,5]. Navafenterol (AZD8871, LAS191351) is an inhaled MABA currently in development for the treatment of COPD. In a phase IIa clinical trial (NCT02971293), once daily administration of navafenterol showed significant and clinically meaningful improvements in lung function-related end points in moderate to severe COPD patients compared with placebo [6]. In a further phase IIa clinical trial (NCT03645434), navafenterol demonstrated improved lung function and a reduction in COPD-related symptoms, similar to the established LAMA/LABA fixed dose combination umeclidinium/vilanterol in patients with moderate to severe COPD. Previous ex vivo studies used human bronchi and guinea pig trachea stimulated by electrical filed stimulation (EFS) to functionally dissect the muscarinic antagonism from the β_2_ adrenergic agonism of navafenterol [7]. In this study, we used precision-cut human lung slices (hPCLS) to functionally dissect the β_2_ adrenergic agonism of navafenterol. Compared to in vivo animal models and ex vivo studies using animal lung tissue, hPCLS yield physiologically relevant findings with translational significance. Instead of EFS stimulation, we used two physiologically relevant contractile agonists (histamine and a thromboxane A_2_ analog) in the presence of the β adrenergic blocker propranolol to functionally isolate the β_2_ adrenergic agonism of navafenterol. Since navafenterol is a known muscarinic antagonist, we used non-muscarinic contractile agonists to examine the β_2_AR agonism of this drug. With a focus on β_2_AR agonism, our findings advance the pharmacological characterization of navafenterol using a physiologically relevant ex vivo platform and supplement other preclinical studies on this investigative drug. 

## 2. Materials and Methods

### 2.1. Human Precision-Cut Lung Slices (hPCLS) 

Precision-cut lung slices were prepared from normal human donor lungs (*n* = 6) as previously described [8]. These samples are exempt from the IRB approval requirement since they are de-identified human tissue. Donor demographics are listed in Table 1. 

### 2.2. Reagents

Navafenterol was provided by AstraZeneca. Propranolol HCl, diluent DMSO, histamine dihydrochloride and thromboxane A_2_ analog (U46619) were obtained from Sigma Aldrich (St. Louis, MO, USA). HAM/F-12 cell culture medium, PBS and media supplements were purchased from Thermo Fisher Scientific (Waltham, MA, USA). 

### 2.3. Reconstitution of Reagents and Exposure Protocol

Navafenterol (10 mM) was reconstituted in DMSO and stored at −20 °C. Propranolol HCl (10 mM) and histamine dihydrochloride (10 mM) were freshly prepared in sterile HAM/F-12 cell culture medium without serum. Thromboxane A_2_ analog (U46619, 10 mg/mL, 28.5 mM) was supplied in methyl acetate and stored at −20 °C. Ten-fold serial dilutions of U46619 or histamine were prepared in HAM/F-12 cell culture medium without serum. Slices were treated with 0.1% DMSO or navafenterol (3, 10, 30, 100 and 300 nM) for 1 h. 

### 2.4. Generation of Concentration Response Curves

Following exposure to navafenterol or vehicle, slices were exposed to incremental concentrations of histamine (10^−10^ M to 10^−4^ M) or U46619 (10^−10^ M to 10^−5^ M). Slices were incubated in each concentration for 5 min in the continued presence of navafenterol or vehicle. In a subset, 10 µM propranolol was co-incubated with each concentration of the contractile agonist to block β_2_AR. The airway lumens were captured and analyzed using Image J as previously described [8]. Briefly, the airway lumen images were captured after incubation with each concentration using an inverted light microscope-linked camera (40× magnification). The luminal areas were measured in each airway with Image J. The change in airway lumen area was calculated as the percentage of baseline area of each airway (percentage bronchoconstriction. Appendix A shows representative images of histamine-induced bronchoconstriction from a single donor). 

### 2.5. Sample Size and Data Analysis

Human PCLS from at least 5 independent lung donors (donor characteristics are provided in Table 1) were used in each experiment. From each donor, 3 slices (technical replicates) were used for each treatment. The mean or mean± SEM of each experimental condition are presented in the graph. The means were statistically compared using GraphPad Prism 9.0, with one-way ANOVA and Dunnett’s test for multigroup comparisons or unpaired, two-tailed Student’s *t*-test for two-group comparisons. The means were considered significantly different if *p* < 0.05.

## 3. Results

### 3.1. Effect of Navafenterol on Histamine-Induced Bronchoconstriction 

Navafenterol (3–300 nM) attenuated histamine-induced bronchoconstriction in a concentration-dependent manner (Figure 1A,B). In the presence of the β_2_AR blocker propranolol (10 µM), navafenterol had little effect on histamine-induced bronchoconstriction at lower (30 and 100 nM) concentrations (Figure 1C,D). However, the highest concentration (300 nM) of navafenterol still attenuated histamine-induced bronchoconstriction in the presence of propranolol. The potency of histamine (p[EC]_50_) was not significantly different in the presence or absence of propranolol (Figure 1E). In slices treated with navafenterol (30 and 100 nM), propranolol increased the potency of histamine compared to the slices not treated with propranolol (Figure 1E). With or without propranolol, histamine-induced maximal contraction (E_max_) was not significantly different in the presence of navafenterol (Figure 1F). 

### 3.2. Effect of Navafenterol on Thromboxane-Induced Bronchoconstriction 

Our findings show that navafenterol had H_1_ receptor antagonism at the highest concentration. To functionally isolate the β_2_AR agonism of navafenterol, we measured the effect of navafenterol on thromboxane A_2_ analog-induced bronchoconstriction. Navafenterol attenuated thromboxane-induced bronchoconstriction only at the highest concentration (300 nM) (Figure 2A,). Blocking of β_2_AR by propranolol completely reversed the inhibitory effect of navafenterol (Figure 2C,D). Thromboxane A_2_ analog-induced bronchoconstriction at the baseline conditions was not affected by propranolol. However, in the presence of navafenterol, propranolol slightly enhanced the potency of thromboxane (Figure 2E). Navafenterol had little effect on thromboxane-induced maximal bronchoconstriction (E_max_) in the presence or absence of propranolol (Figure 2F).

## 4. Discussion

Navafenterol is a novel, single-molecule, dual pharmacology bronchodilator combining muscarinic (M_3_) cholinergic receptor antagonist and β_2_AR agonist functions [7]. Previous pharmacological characterizations of navafenterol were performed in isolated human bronchi and guinea pig tracheal rings using electrical field stimulation (EFS). Studies also focused on guinea pig and canine models to assess the bronchoprotective and off-target effects of navafenterol. The objective of the current study is to functionally dissociate the β_2_AR agonism of navafenterol from its reported antagonism towards histamine (H_1_) receptors. 

Human PCLS is an innovative ex vivo platform to characterize investigational drugs, toxicants and infectious agents. The direct physiological measurements from small airways in hPCLS have immense physiological and translational values compared to other preclinical models. The current study is yet another piece of evidence that demonstrates the application of hPCLS in the pharmacological characterization of a drug targeting distinct receptors in human airways. 

A number of investigational drugs with combined muscarinic antagonist and β_2_ adrenergic agonist activities have been developed [3,5,9]. These small molecules are combined antagonist and agonist entities connected by a chemical linker with the structure of the linker influencing the balance between muscarinic antagonist and β_2_AR agonist functions. In isolated human bronchi, navafenterol showed more dominant muscarinic antagonism than β_2_AR agonism compared to batefenterol (GSK961081), another MABA [7]. The same study identified navafenterol to have moderate affinity for histamine H_1_ receptors (IC_50_ = 85 nM, p[IC]_50_ = 7.1). Our findings support this observation by showing that 300 nM of navafenterol retains the inhibitory effect on bronchoconstriction in the presence of propranolol with a similar measure of compound affinity (pA_2_) of 7.5 (Table 2). 

Thromboxane is one of the several prostanoids with roles in airway inflammation and hyperresponsiveness (reviewed in [10]). Acting through thromboxane prostanoid (TP) receptors, TxA_2_ signals through G_αq/11_, mobilizing cytosolic Ca^2+^ and eliciting bronchoconstriction. Thromboxane A_2_ analog-induced bronchoconstriction was used to further demonstrate the β_2_AR agonism of navafenterol. In contrast to histamine-induced bronchoconstriction, the inhibitory effect of navafenterol on TxA_2_ analog-induced airway narrowing was modest, showing significant inhibition only at the highest concentration of the drug. However, the complete reversal of that navafenterol inhibition in the presence of propranolol suggests that thromboxane prostanoid (TP) receptors are not antagonized by navafenterol.

In summary, we have functionally distinguished the histamine antagonism of navafenterol from its β_2_AR agonism using two distinct contractile agonists in human small airways. These findings demonstrate that the bronchoprotective effect of navafenterol in human small airways is primarily mediated through β_2_AR agonism (summarized in Figure 3). Further studies are required to determine if the H_1_ antagonism seen with navafenterol may contribute to a therapeutically meaningful bronchoprotective effect in diseases where histamine may play a role.

## Figures and Tables

**Figure 1 cells-12-00240-f001:**
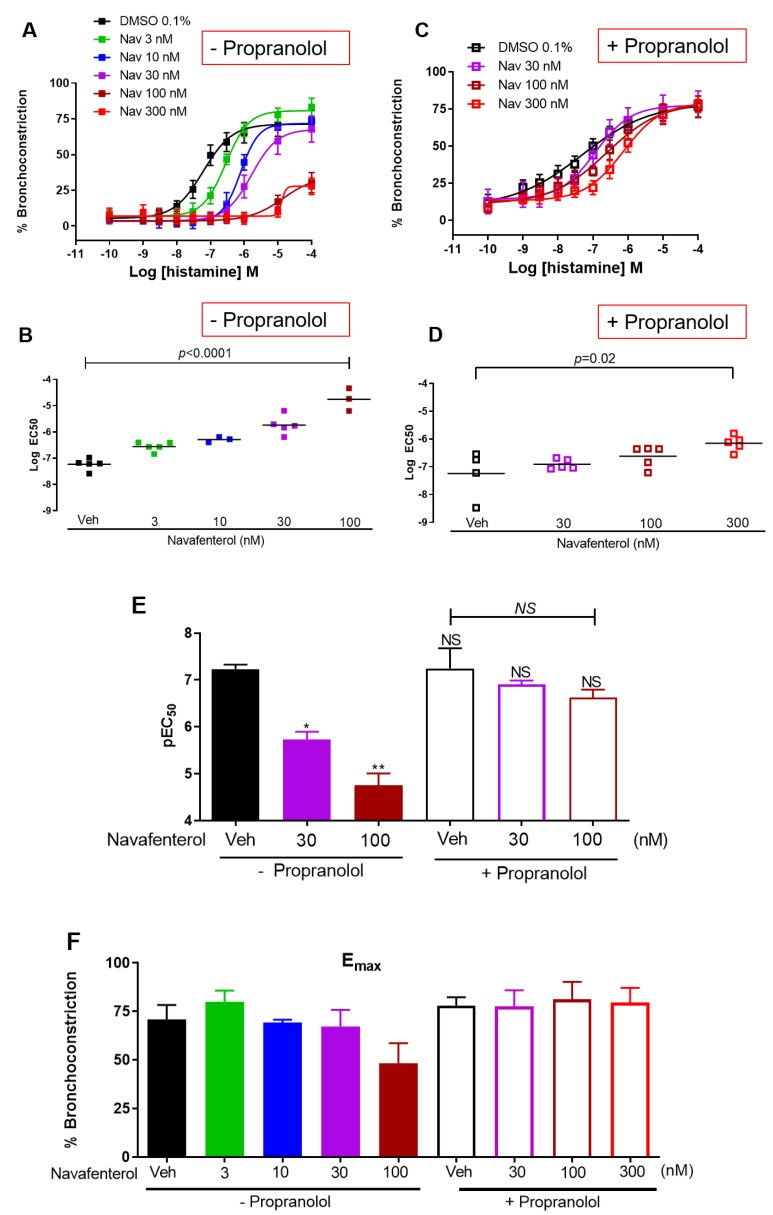
Effect of navafenterol (Nav) on histamine-induced bronchoconstriction. (**A**,**B**) Navafenterol attenuates histamine-induced bronchoconstriction in a concentration-dependent manner (*n* = 5 donors, *p* < 0.0001, one-way ANOVA, all conditions compared to Veh). (**C**,**D**) In the presence of propranolol (10 µM), the inhibitory effect of navafenterol on histamine-induced bronchoconstriction is reversed, except at the highest (300 nM) concentration (*n* = 5 donors, *p* < 0.05, one-way ANOVA, all conditions compared to Veh). (**E**) Propranolol has little effect on histamine potency at the baseline (Veh), while significantly increasing histamine potency in navafenterol-treated airways (*n* = 3 to 5 donors, * *p* = 0.0003, ** *p* < 0.0001, NS- not significant, compared to Veh without propranolol (black bar); *NS* under black line: not significant compared to Veh with propranolol (white bar); one-way ANOVA with Tukey’s multi-group comparison test. Data points for E were obtained from B and D). (**F**) Maximal contraction (E_max_) showed a reduced trend in the presence of 100 nM Navafenterol and the absence of propranolol (*n* = 3 to 5 donors, one-way ANOVA with Tukey’s multiple comparison test, none of the groups were statistically significant).

**Figure 2 cells-12-00240-f002:**
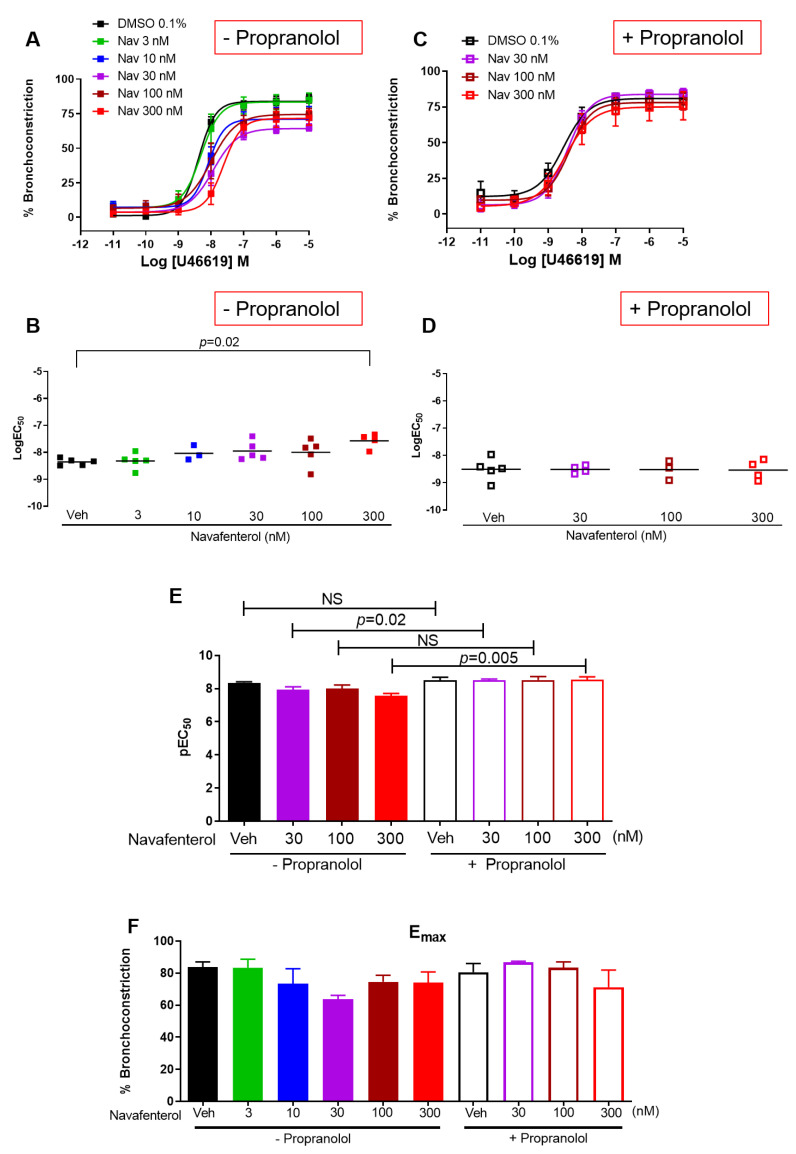
Effect of navafenterol (Nav) on thromboxane-induced bronchoconstriction. (**A**,**B**) The highest concentration of navafenterol (300 nM) attenuates TxA_2_-induced bronchoconstriction (*n* = 5 donors, *p* < 0.05, one-way ANOVA, all conditions compared to Veh). (**C**,**D**) In the presence of propranolol (10 µM), the inhibitory effect of navafenterol on TxA_2_-induced bronchoconstriction is reversed (*n* = 3 to 5 donors, *p* < 0.05, one-way ANOVA, all conditions compared to Veh). (**E**) Propranolol has little effect on TxA_2_ potency at the baseline (Veh) while significantly increasing TxA_2_ potency in navafenterol-treated airways (*n* = 3 to 5 donors, *p* < 0.05, unpaired Student’s t-test comparing each pair as indicated in the graph). (**F**) Navafenterol has little effect on TxA_2_-induced maximal contraction (E_max_) in the presence or absence of propranolol (*n* = 3 to 5 donors, one-way ANOVA with Tukey’s multi-group comparison test).

**Figure 3 cells-12-00240-f003:**
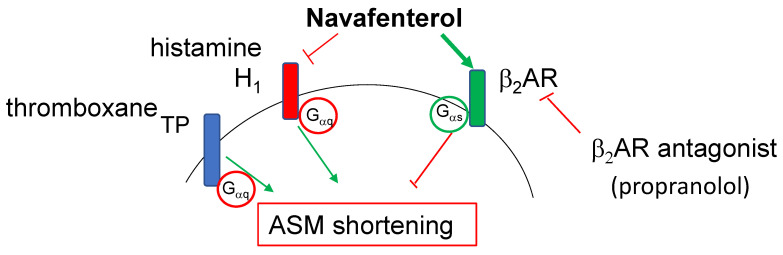
Schematic model of dual action for navafenterol (AZD8871) in human small airways. Navafenterol mediates bronchoprotective effect primarily through its agonism (thick green arrow) towards β_2_AR. At the highest concentration, H_1_ antagonism (thin red line, flat end) contributes to the bronchoprotective effect of navafenterol. Navafenterol has little effect on thromboxane prostanoid (TP) receptor-mediated bronchoconstriction.

**Table 1 cells-12-00240-t001:** Characteristics of human lung donors (*n* = 6). Qualitative and quantitative characteristics of 6 lung donors used in the study are listed. In total, 5 donors were used in each histamine and thromboxane experiment (4 donors were used in both experiments). The cause of mortality for these donors ranged from head trauma to cerebrovascular disease with no reported lung pathology.

	Mean Age (SD)	38.8 (8.7)
Sex	M	04
	F	02
Race	Caucasian	03
	Black	02
	Hispanic	01
BMI, Kgm^−2^ (SD)		32.5 (8.3)

**Table 2 cells-12-00240-t002:** Potency (p[EC]_50_) of histamine in the presence or absence of propranolol and navafenterol. The p[EC]_50_ values from histamine cumulative concentration–response curves were used to calculate pA_2_ values for navafenterol in the presence of propranolol-induced β_2_AR blockade. Compared to vehicle control, the p[EC]_50_ of histamine in the presence of 300 nM navafenterol was significantly decreased (* *p* < 0.05, one-way ANOVA with Dunnett’s multiple comparison test, *n* = 4 donors). This was used to determine a pA_2_ value for navafenterol at the histamine receptor of 7.5.

Treatment	DMSO 0.1% p[EC]_50_	Histamine p[EC]_50_ in the Presence of Propranolol		
		30 nM Navafenterol	100 nM Navafenterol	300 nM Navafenterol
**Mean ± SEM**	7.25 ± 0.43	6.95 ± 0.09	6.69 ± 0.20	* 6.16 ± 0.16

## Data Availability

Data is contained within the article and Appendix A. Original numerical data will be made available upon reasonable request.

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
