# Peer review of "The Bronchoprotective Effects of Dual Pharmacology, Muscarinic Receptor Antagonist and β2 Adrenergic Receptor Agonist Navafenterol in Human Small Airways"

_cells, 2023, doi:10.3390/cells12020240_

Round 1
Reviewer 1 Report
This is a concise study with significant findings. The experimental design is straightforward and carefully implemented. The major conclusion that the bronchoprotective effect of navafenterol in human small airways is primarily mediated through beta2AR agonism is supported by the data.
Minor and specific
1) Line 16: Typo, “is a single molecule, …”
2) lines 20-21: “Navafenterol pre-treatment significantly attenuated histamine-induced bronchoconstriction and β2AR antagonist propranolol reversed this inhibitory effect”. This conclusion is based on the data shown in Fig. 1. In Fig. 1E, in the presence of propranolol (white bars) there should be a one-way ANOVA to show whether the decrease due to the increase in navafenterol is significant. If it is, then propranolol did not completely reverse the effect of navafenterol, and the conclusion should be that propranolol partially reversed the inhibitory effect.
3) lines 101 and 147: “β” is missing.
4) The is a long description for Table 2 (lines 98-104). Some of the description can be moved to the main text.
Author Response
We thank the reviewer for the insights. Our responses are below:
1) Line 16: Typo, “is a single molecule, …”
Thanks. We have corrected this.
2) lines 20-21: “Navafenterol pre-treatment significantly attenuated histamine-induced bronchoconstriction and β2AR antagonist propranolol reversed this inhibitory effect”. This conclusion is based on the data shown in Fig. 1. In Fig. 1E, in the presence of propranolol (white bars) there should be a one-way ANOVA to show whether the decrease due to the increase in navafenterol is significant. If it is, then propranolol did not completely reverse the effect of navafenterol, and the conclusion should be that propranolol partially reversed the inhibitory effect.
We have updated the analyses of Figure 1E according to the reviewer’s suggestion. As indicated in the revised figure and legend, One-Way ANOVA showed no significant changes in 30 or 100 nM Navafenterol in the presence of propranolol (White bar). This analysis supports our original conclusion that propranolol reverses the inhibitory effect of navafenterol.
3) lines 101 and 147: “β” is missing.
We have corrected these errors.
4) The is a long description for Table 2 (lines 98-104). Some of the description can be moved to
We have shortened the title to “Potency (p[EC]50) of histamine in the presence or absence of propranolol and navafenterol”
Reviewer 2 Report
General Comments:
This manuscript, authored by Jude JA et al., is well-written and provides interesting findings related to the bronchoprotective effects of navafenterol in human airways. However, minor points need to be improved for a better understanding of this manuscript. Specific comments are included below.
Specific Comments:
1. Authors show in figure 1B the p[EC]50 of all navafenterol concentrations used. The navafenterol 100 nM p[EC]50 was obtained from the concentration-response curve to histamine in figure 1A; however, this curve, in the presence of 100 nM navafenterol is incomplete. Therefore the p[EC]50 cannot be obtained unless it has been extrapolated. Therefore, to acquire this parameter, authors must complete the curve as in figure 1B. Moreover, in figure 1A, the authors should analyze the maximum contraction obtained in each curve to determine if there are differences in this response.
2. In figure 2A the authors should analyze if there are statistical differences in the maximum response between the groups.
Author Response
We thank the reviewer for the insights. The following are our responses:
- Authors show in figure 1B the p[EC]50 of all navafenterol concentrations used. The navafenterol 100 nM p[EC]50 was obtained from the concentration-response curve to histamine in figure 1A; however, this curve, in the presence of 100 nM navafenterol is incomplete. Therefore the p[EC]50 cannot be obtained unless it has been extrapolated. Therefore, to acquire this parameter, authors must complete the curve as in figure 1B. Moreover, in figure 1A, the authors should analyze the maximum contraction obtained in each curve to determine if there are differences in this response.
We thank the reviewer for the observations.
- On completeness of the 100 nM Nav curve: This is the complete concentration-response curve and the significant flattening of the curve is due to the presence of 100 nM Nav (similarly, 300 nM Nav flattens the CRC as well). The semi-log curves were analyzed with a variable slope (i.e: Hill slope not fixed at 1.0) concentration -response curve fitting in GraphPad Prism 9.0, that calculates the Log EC50 used in Figure 1B. Since the curve shape in 100 nM Nav appears to be the biological effects of Nav, it is not feasible to analyze these curves by any means other than variable slope curve fitting and extrapolation.
- Emax: We have updated the Figure 1 with histamine Emax graph in Figure 1F. Note that the Emax values were the extrapolated values obtained from variable slope curve-fitting.
- In figure 2A the authors should analyze if there are statistical differences in the maximum response between the groups.
We have included the new Figure 2F showing maximal contraction (Emax) calculated by variable slope curve-fitting. There were no significant differences in Emax between the conditions.
Reviewer 3 Report
The manuscript by Jude et al. investigated the bronchoprotective effect of navafenterol against two non-muscarinic contractile agonists, histamine and thromboxane 19 A2 (TxA2) analog (U46619) in precision-cut human lung slices (hPCLS).
The manuscript is well written and the results are interesting, anyway to me it is a little bit difficult to understand how the % of bronchoconstriction was obtained since no images of airways are shown. In my opinion, before acceptance, it would be appropriate to show some representative images of the controls and treated samples in order to appreciate the difference in terms of bronchoconstriction.
Author Response
We thank the reviewer for the comment. Here is our response:
The manuscript is well written and the results are interesting, anyway to me it is a little bit difficult to understand how the % of bronchoconstriction was obtained since no images of airways are shown. In my opinion, before acceptance, it would be appropriate to show some representative images of the controls and treated samples in order to appreciate the difference in terms of bronchoconstriction.
We thank the reviewer. We have included a supplementary Figure (Figure S1) showing 4 representative airways imaged after exposure to histamine (10-10 M to 10-4 M) in the presence of Navafenterol 100 nM -/+ 10 uM propranolol. Essentially, a single airway, following experimental exposures, is treated with incremental concentrations of histamine (5 min) and imaged under a light microscope at 40X magnification. The captured images are analyzed by measuring the airway lumen area (white space) using Image J software. The % reduction in the lumen area in each individual airway is calculated by normalizing the luminal areas to that airway’s baseline (BL). The % change in airway luminal area is called “% bronchoconstriction”. Thus, % bronchoconstriction at baseline condition in each airway will be 0 %. Because the luminal area change at each concentration of histamine is normalized to the baseline of the same airway, variations in actual size and shape of the airways used in different experimental groups are eliminated from the analysis.
Round 2
Reviewer 3 Report
In my opinion, the inclusion of the supplementary figure to describe bronchoconstriction has made the work more complete and understandable. The manuscript can be accepted for publication in its present form.